# Insulin Resistance Is the Main Characteristic of Metabolically Unhealthy Obesity (MUO) Associated with NASH in Patients Undergoing Bariatric Surgery

**DOI:** 10.3390/biomedicines11061595

**Published:** 2023-05-31

**Authors:** Sophia M. Schmitz, Sebastian Storms, Alexander Koch, Christine Stier, Andreas Kroh, Karl P. Rheinwalt, Sandra Schipper, Karim Hamesch, Tom F. Ulmer, Ulf P. Neumann, Patrick H. Alizai

**Affiliations:** 1Department of General-, Visceral- and Transplantation Surgery, RWTH Aachen University Hospital, Pauwelsstr. 30, 52074 Aachen, Germanypalizai@ukaachen.de (P.H.A.); 2Department of Gastroenterology, Digestive Diseases and Intensive Care Medicine, RWTH Aachen University Hospital, Pauwelsstr. 30, 52074 Aachen, Germany; 3Department of Internal Medicine, Division of Endocrinology and Diabetes, University Hospital Wuerzburg, 97080 Wuerzburg, Germany; 4Department of General-, Visceral-, Transplantation, Vascular and Pediatric Surgery, University Hospital Wuerzburg, 97080 Wuerzburg, Germany; 5Department of Surgical Endoscopy, Sana Hospital Huerth, 50354 Huerth, Germany; 6Department of Bariatric, Metabolic and Plastic Surgery, St. Franziskus-Hospital, Schönsteinstr. 63, 50825 Cologne, Germany; 7Department of Surgery, Maastricht University Medical Center, P. Debyelaan 25, 6229 HX Maastricht, The Netherlands; 8Gemeinschaftskrankenhaus Bonn, Klinik für Allgemein- und Viszeralchirurgie, 53113 Bonn, Germany

**Keywords:** NAFLD, metabolically unhealthy obesity, obesity surgery, insulin resistance

## Abstract

(1) Background: Metabolically healthy obesity (MHO) is a concept that applies to obese patients without any elements of metabolic syndrome (metS). In turn, metabolically unhealthy obesity (MUO) defines the presence of elements of metS in obese patients. The components of MUO can be divided into subgroups regarding the elements of inflammation, lipid and glucose metabolism and cardiovascular disease. MUO patients appear to be at greater risk of developing non-alcoholic fatty liver disease (NAFLD) and non-alcoholic steatohepatitis (NASH) compared to MHO patients. The aim of this study was to evaluate the influence of different MUO components on NAFLD and NASH in patients with morbid obesity undergoing bariatric surgery. (2) Methods: 141 patients undergoing bariatric surgery from September 2015 and October 2021 at RWTH Aachen university hospital (Germany) were included. Patients were evaluated pre-operatively for characteristics of metS and MUO (HbA1c, HOMA, CRP, BMI, fasting glucose, LDL, TG, HDL and the presence of arterial hypertension). Intraoperatively, a liver biopsy was taken from the left liver lobe and evaluated for the presence of NAFLD or NASH. In ordinal regression analyses, different factors were evaluated for their influence on NAFLD and NASH. (3) Results: Mean BMI of the patients was 52.3 kg/m^2
^ (36–74.8, SD 8.4). Together, the parameters HbA1c, HOMA, CRP, BMI, fasting glucose, LDL, TG, HDL and the presence of arterial hypertension accounted for a significant amount of variance in the outcome, with a likelihood ratio of χ^2^ (9) = 41.547, *p* < 0.001, for predicting the presence of NASH. Only HOMA was an independent predictor of NASH (B = 0.102, SE = 0.0373, *p* = 0.007). Evaluation of steatosis showed a similar trend (likelihood ratio χ^2^ (9) = 40.272, *p* < 0.001). Independent predictors of steatosis were HbA1c (B = 0.833, SE = 0.343, *p* = 0.015) and HOMA (B = 0.136, SE = 0.039, *p* < 0.001). (4) Conclusions: The above-mentioned model, including components of MUO, was significant for diagnosing NASH in patients with morbid obesity undergoing bariatric surgery. Out of the different subitems, HOMA independently predicted the presence of NASH and steatosis, while HbA1c independently predicted steatosis and fibrosis. Taken together, the parameter of glucose metabolism appears to be more accurate for the prediction of NASH than the parameters of lipid metabolism, inflammation or the presence of cardiovascular disease.

## 1. Introduction

Obesity itself and obesity-related comorbidities are on the rise worldwide. In Europe, more than 20% of the adult population is obese [1]. Obesity is a risk factor for other comorbidities, such as cardiovascular disease, diabetes mellitus (T2DM) and non-alcoholic fatty liver disease (NAFLD). However, some individuals appear to be at a higher risk than others. In recent years, the concepts of metabolically healthy obesity (MHO) and metabolically unhealthy obesity (MUO) have gained attention. So far, there is no universal definition of MHO. All concepts combine the presence of obesity (BMI > 30 kg/m^2^) with the absence or minimal presence of parameters of metabolic syndrome (metS) [2,3,4,5,6,7,8,9]. Particularly well-established criteria proposed by Wildman et al. and later modified by Stefan are frequently referred to as “modified Wildman criteria” [2,10]. A higher risk for NAFLD with increasing degree of obesity is known, while the influence of MUO on the presence of NAFLD is not yet fully understood [11,12]. Some studies even point towards an increased risk of NAFLD depending on metabolic status, independent from visceral fat [4].

Nowadays, NAFLD is the main cause of chronic liver disease in most Western countries [13]. Additionally, in patients with NAFLD, the presence of metS has been reported to be associated with higher mortality, while metabolically normal patients with NAFLD appear to be at a comparable risk to patients without NAFLD [14]. Therefore, deep understanding of the pathophysiology and therapeutic options is of utmost importance. NAFLD itself triggers other factors of metS and might not only be a consequence but also a cause of metS [11,15,16,17]. Furthermore, progression of MHO individuals towards MUO might be facilitated by the presence of NAFLD [18]. However, even in MHO patients, NAFLD is far from uncommon [3,5,6]. In patients with obesity, a growing body of evidence has demonstrated that bariatric surgery leads to improvement, and sometimes even complete resolution, of NAFLD and NASH as well as liver fibrosis [19,20,21,22,23].

Most patients undergoing bariatric surgery show characteristics of MUO [3,9]. Similar to improvement of NAFLD, bariatric surgery leads to an improvement of metabolic markers in both MHO and MUO patients [9]. One year after bariatric surgery, a shift from MUO to MHO in 87% of patients has been described [9]. MHO, at baseline, might lead to higher weight loss after bariatric surgery [24]. To date, however, much uncertainty still exists about the relationship and the importance of the different parameters of MUO with respect to occurrence and progression of comorbidities.

The aim of this study was to assess the influence of the different subitems of MUO on NAFLD and NASH in patients with obesity.

## 2. Materials and Methods

Patients with morbid obesity that underwent bariatric surgery from September 2015 to October 2021 at our university’s obesity surgery center (RWTH Aachen University Hospital, Aachen, Germany) were included in this retrospective study. Patients with a history of alcohol consumption (>20 mg/day for women and >40 mg/day for men) were excluded from the study. Sex, height, weight, body mass index (BMI) and presence of the comorbidities T2DM and arterial hypertension were assessed. Blood was drawn after fasting overnight during the two weeks prior to the operation and the following blood values were determined: glycohemoglobin (HbA1c), homeostatic model assessment for insulin resistance (HOMA-IR), C-reactive protein (CRP), fasting glucose, low-density lipoprotein (LDL), triglycerides (TG) and high-density lipoprotein (HDL). Intraoperatively, a liver biopsy was taken from the left liver lobe and assessed for the presence of NAFLD or NASH by experienced hepato-pathologists by applying the NAFLD activity score (NAS), as reported by Kleiner et al. [25].

For diagnosis of MUO, the following parameters were used: (a) systolic blood pressure (BP) ≥ 130 mmHg, diastolic BP ≥ 85 mmHg, use of oral antihypertensive medication or previous diagnosis of hypertension; (b) triglycerides ≥ 150 mg/dL or use of lipid-lowering medications; (c) HDL cholesterol < 40 mg/dL (men) or <50 mg/dL (women); (d) fasting glucose ≥ 110 mg/dL, use of hypoglycemic agents or previous diagnosis of diabetes mellitus; (e) HOMA index > 2.5; and (f) CRP > 5 mg/dL (criteria modified according to Wildman et al. and Stefan et al. [2,10]). This study was approved by the institutional review board of RWTH Aachen University Hospital (EK# 196/22) and carried out in accordance with the ethical standards of the Helsinki Declaration and its further amendments.

### Statistical Analysis

Statistical analysis was performed with IBM SPSS Statistics v.22 and GraphPad Prism 9. *p* < 0.05 was considered to represent a statistically significant difference. Outliers above and below 2 standard deviations (SD) were excluded from further analysis. Ordinal regression analysis was used to assess the influence of the different MUO parameters on the presence of NAFLD or NASH and the subitems steatosis, activity and fibrosis. BMI, TG, HDL, LDL, glucose, HbA1c, HOMA and CRP were included as independent numerical variables, while the presence of arterial hypertension was included as an independent categorical variable.

## 3. Results

A total of 141 patients with a mean BMI of 52.3 kg/m^2^ were included in this study. The mean age was 43.3 years, and 101 patients (71.6%) were female. A total of 55 patients had an NAFLD activity score (NAS) of 0–2 and therefore showed no signs of NASH (39%), 67 patients had a score of 3 or 4 and were therefore classified as borderline (47.5%) and 19 patients had a score of 5 or more and were therefore classified as definite NASH (13.5%). For information about biometric parameters and blood values, see Table 1.

Out of the criteria for diagnosing MUO, only one patient presented without positive criteria for MUO (0.7%). A total of 19 patients (13.5%) presented with one or two positive criteria, while 121 patients (85.8%) presented with three or more positive criteria for diagnosis of MUO. A total of 46 out of 141 patients (32.6%) had a diagnosis of diabetes mellitus. A total of 25 patients were treated with metformin (17.7%), 3 patients were treated with SGLT2-Inhibitors (2.1%), 1 patient was treated with thiazolidinediones (0.7%), 1 patient was treated with dipeptidyl peptidase-4 inhibitor (0.7%) and 3 patients were treated with GLP1-agonists (2.1%). The mean values for the tested variables stratified by number of positive MUO criteria are indicated in Table 2.

The proportion of positive MUO criteria in relation to the NAFLD activity score is shown in Figure 1.

The statistical ordinal regression model, including HbA1c, HOMA, CRP, BMI, fasting glucose, LDL, TG, HDL and the presence of arterial hypertension, was significant for predicting NASH (likelihood ratio χ^2^ (9) = 41.547, *p* < 0.001). HOMA was the only independent predictor for NASH (B = 0.102, SE = 0.0373, *p* = 0.007) (see Table 3).

For the subitem steatosis, the likelihood ratio for the model was χ^2^ (9) = 40.272, *p* < 0.001. HbA1c (B = 0.833, SE = 0.343, *p* = 0.015) and HOMA (B = 0.136, SE = 0.039, *p* < 0.001) independently predicted steatosis. The proportion of positive MUO criteria in relation to steatosis is shown in Figure 2. An overview of the model is presented in Table 4.

The subitem activity could not be predicted with the model used (χ^2^ (9) = 14.013, *p* = 0.122), while for fibrosis, the model met statistical significance (χ^2^ (9) = 24.515, *p* = 0.004), with HbA1c as an independent predictor for fibrosis (B = 1.006, SE 0.349, *p* = 0.004). See Table 5 and Table 6 for an overview of the independent parameters and Figure 3 and Figure 4 for the number of positive MUO criteria in relation to degree of activity and fibrosis, respectively.

## 4. Discussion

So far, there are few data on the influence of the different parameters of MUO on the prevalence of the obesity-associated comorbidities NAFLD and NASH. Different definitions and concepts of MUO further complicate the clarification of the underlying pathophysiological processes. As a consequence, data on the prevalence of MUO and MHO in patients undergoing bariatric surgery vary widely. For MHO, Lee et al. found a prevalence of nearly 30% with a rather broad definition, based only on the presence of arterial hypertension and diabetes mellitus [3]. When elevated fasting triglycerides were included in the analysis, the prevalence of MHO was found to be only 17% [3]. Goday et al. found a prevalence of 18% of MHO in patients undergoing bariatric surgery, applying the definition by Wildman et al., without including CRP [9]. This widely used definition for diagnosis of MUO is based on elevated blood pressure, elevated triglyceride level, decreased HDL-C level, elevated glucose level, insulin resistance and systemic inflammation [2]. Other studies base their definition of MHO on the definition of metS, with less than two out of elevated fasting glucose, elevated blood pressure, reduced HDL and elevated triglyceride levels, without mentioning HOMA-IR and CRP [4,7]. These criteria were used in two other studies that required the strict absence of any of these for diagnosis of MHO. In this manner, Haskins et al. found a prevalence of 7% while including patients with a BMI > 30 kg/m^2^, while Frey et al. included patients with a BMI > 35 kg/m^2^ and found a prevalence of 18% for MHO [5,6]. Genua et al. published a retrospective analysis of patients operated on in their institution in 2021 and found a prevalence of MHO of 19%, applying the criteria absence of T2DM or atherogenic dyslipidemia, absence of treatment with hypoglycemic treatment or treatment with fibrates, low plasma glucose, low HbA1c, low triglycerides and high HDL [24]. In comparison, in our study, only one patient (0.7%) did not show any parameters of MUO, while 4 patients (2.8%) had one characteristic and 15 patients (10.6%) showed 2 characteristics of MUO via the definition of Wildman et al. [2]. Therefore, the prevalence of MUO in patients undergoing bariatric surgery appears to be higher in our cohort than in previously reported studies. One reason might be the high mean BMI of 52.3 kg/m^2^ of patients included in our study, which is mainly due to restrictions and impediments of bariatric surgery in Germany, where operations are performed later than in other countries.

However, patients’ BMI is only one parameter in the complex relationship between MUO and NAFLD. In line with this, Chen et al. report a higher risk of development of NAFLD in MUO when compared to MHO, but also to metabolically unhealthy non-obese (MUNO) and metabolically healthy non-obese (MHNO) patients. In this study, obesity and metabolically unhealthy status were both considered risk factors for NAFLD, independent of visceral fat. However, diagnosis of NAFLD in this population-based study was based on ultrasonography and was not biopsy-proven [4].

Likewise, Huh et al. found an increasing risk of liver steatosis measured by transient elastography for health status and obesity assessed with BMI. The lowest grade of steatosis was found in MHNO, followed by MUNO. The highest grades of steatosis were found in MHO and MUO [7]. In a multivariate analysis, sex, BMI, HDL, fasting glucose and ALT were significant for controlled attenuation parameters via transient elastography. However, insulin resistance, as reflected by HOMA-IR, was not taken into account in their study. In line with these results, Kotronen et al. reported an increased amount of liver fat content, measured by proton magnetic resonance spectroscopy, in patients with metabolic syndrome. The difference remained significant after adjusting for age, gender and BMI [26]. Tutunchi et al. reported on an increased risk of progression of NAFLD depending on the amount of body fat but regardless of metabolic status [27]. In our study, the used ordinal regression model, including the parameters of weight, glucose and fat metabolism, insulin resistance and inflammation, was significant for predicting NASH in patients with morbid obesity. The only independently significant parameter was HOMA-IR. Therefore, insulin resistance is of paramount importance in connecting NAFLD and metabolic syndrome in a reciprocal manner [16]. In line with this and as a possible pathophysiological explanation, Yki-Jarvinen points out that the ability of the liver to produce two key components of metS (VLDL, containing triglycerides, and fasting plasma glucose) is usually suppressed by insulin. In insulin resistance, as is the case in NAFLD, the liver, therefore, starts overproducing glucose and VLDL [17]. Similarly, Kotronen et al. report a correlation of fasting serum insulin and C-peptide with fat content of the liver [26].

Insulin resistance has been named the key factor of MUO before [24]. Frey et al. compared MHO with MUO patients undergoing bariatric surgery and found significant differences in CRP, HbA1c and HOMA-IR in a multivariate analysis. For severe steatosis in both MHO and MUO patients, BMI and ALT were significant in the multivariate analysis [5]. In our study, we found a clear increase in degree of steatosis with an increasing number of parameters of MUO. In ordinal regression analysis, HOMA-IR and HbA1c were independent predictors of steatosis. Furthermore, HbA1c was an independent predictor of fibrosis, the most important long-term-outcome parameter in NAFLD and NASH.

With regard to our study, we can only speculate whether MUO is the reason for NAFLD, or vice versa. Most likely, both entities are intertwined in a reciprocal manner. However, when systematically assessing the risk factors of MUO, a clear tendency towards the importance of the parameters of insulin resistance and glucose metabolism rather than inflammation, lipid metabolism or cardiovascular disease can be seen. At the same time, this is the main limitation of this study. Due to the nature of the study design, we cannot determine what is the cause and what is the effect in the interdependence of MUO and NAFLD/NASH. Another limitation is the rather small number of included patients. As mechanisms linking MUO and NAFLD are not entirely known and due to the study design, there might be unknown confounding factors that could not be accounted for. Hence, further randomized, controlled trials with interventions concerning glucose metabolism, inflammation, lipid metabolism and optimization of cardiovascular risk factors should confirm our findings and demonstrate therapeutic options based on the different targets.

## 5. Conclusions

With the rising worldwide prevalence of obesity in recent years, the concept of metabolically healthy obesity (MHO) has gained attention. The definition of MHO requires the absence or minimal presence of the factors of metabolic syndrome in obese patients. However, the concept lacks a definition of universal validity. In obese patients undergoing bariatric surgery, NAFLD and NASH are common comorbidities. In our study, we used an ordinal regression model, including the parameters of weight, glucose and fat metabolism, insulin resistance and inflammation, to predict NASH in patients with morbid obesity undergoing bariatric surgery. The model itself was statistically significant for prediction of NASH, while the only independently significant parameter was HOMA-IR. We therefore hypothesize that insulin resistance is of paramount importance in connecting NAFLD and metabolic syndrome in a reciprocal manner.

## Figures and Tables

**Figure 1 biomedicines-11-01595-f001:**
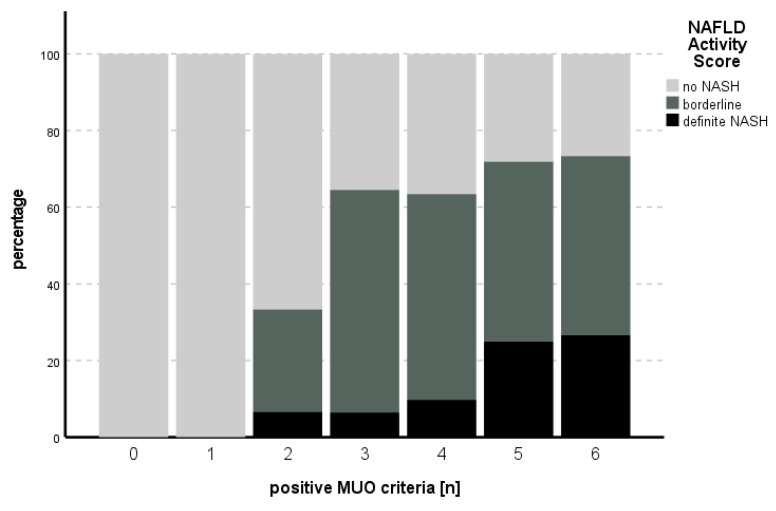
Number of positive MUO criteria in comparison to the percentage of NAS stage. An increasing percentage of definite NASH could be seen with increasing numbers of positive MUO criteria.

**Figure 2 biomedicines-11-01595-f002:**
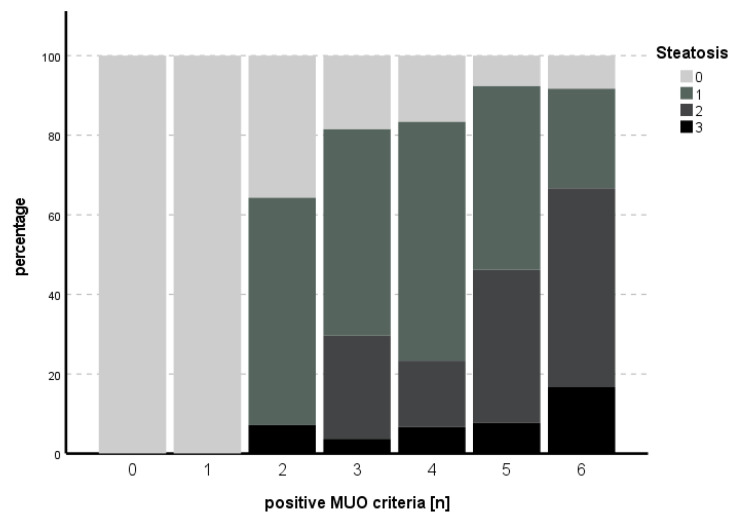
Number of positive MUO criteria in comparison to degree of steatosis. An increasing percentage of higher degrees of steatosis could be seen with increasing numbers of positive MUO criteria.

**Figure 3 biomedicines-11-01595-f003:**
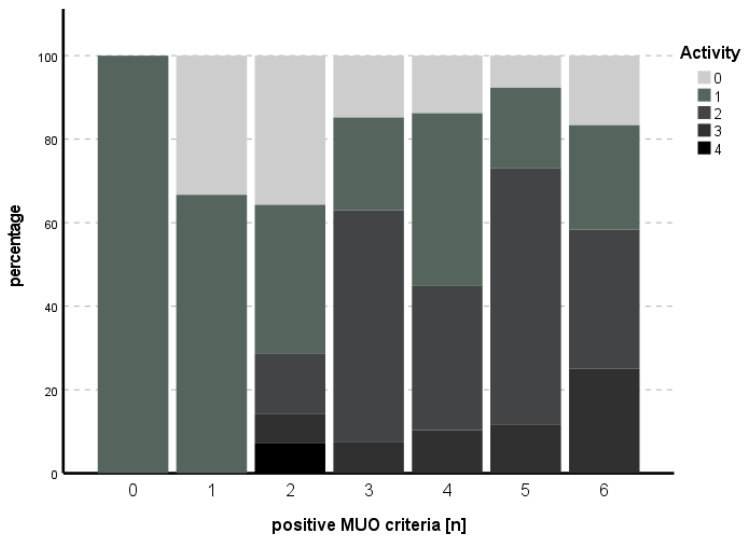
Number of positive MUO criteria in comparison to degree of activity. An increasing percentage of higher degrees of activity could be seen with increasing numbers of positive MUO criteria.

**Figure 4 biomedicines-11-01595-f004:**
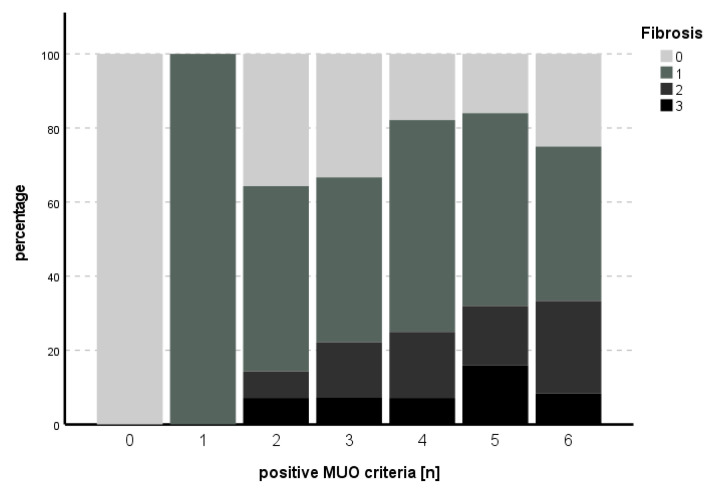
Number of positive MUO criteria in comparison to degree of fibrosis. An increasing percentage of higher degrees of fibrosis could be seen with increasing numbers of positive MUO criteria for up to five criteria, while this trend could not be established in the case of six positive MUO criteria.

**Table 1 biomedicines-11-01595-t001:** General information on the study population. Abbreviations: Art. HT: presence of arterial hypertension; BMI: body mass index; TG: triglyceride; HDL: high-density lipoprotein; LDL: low-density lipoprotein; HbA1c: hemoglobin A1c; HOMA: homeostasis model assessment; CRP: C-reactive protein; SD: standard deviation.

	N	Mean	SD	Number of Positive Criteria for MUO
BMI [kg/m^2^]	141	52.3	8.4	-
Age [years]	141	43.3	10.5	-
Female Sex	101/141	-	-	-
TG [mg/dL]	141	163.0	93.8	58 (41%)
HDL [mg/dL]	141	45.4	11.1	84 (59.6%)
LDL [mg/dL]	141	129.6	29.5	-
Glucose [mg/dL]	141	116.4	51.6	79 (56%)
HbA1c [%]	141	6.2	1.5	-
HOMA Index	141	10.1	9.2	131 (93%)
CRP [mg/dL]	141	11.4	8.4	110 (78%)
Art. HT [yes]	85/141	-	-	85 (60%)

**Table 2 biomedicines-11-01595-t002:** Means of the MUO parameters indicated by number of positive criteria for diagnosis of MUO. Abbreviations: Art. HT: arterial hypertension; BMI: body mass index; TG: triglyceride; HDL: high-density lipoprotein; LDL: low-density lipoprotein; HbA1c: hemoglobin A1c; HOMA: homeostasis model assessment; CRP: C-reactive protein; SD: standard deviation.

	Number of Positive Criteria for Diagnosis of MUO																				
	0			1			2			3			4			5			6		
	n	Mean	SD	n	Mean	SD	n	Mean	SD	n	Mean	SD	n	Mean	SD	n	Mean	SD	n	Mean	SD
TG	1	65.00	-	4	86.00	36.67	15	105.80	29.10	32	134.94	63.97	41	149.98	67.99	33	179.37	61.30	15	267.40	98.82
HDL	1	61.00	-	4	64.50	15.72	15	46.47	8.82	32	49.81	11.77	41	44.29	11.47	33	42.09	7.72	15	39.47	6.21
LDL	1	100.00	-	4	112.25	13.57	15	120.47	30.31	32	133.59	30.80	41	130.73	29.32	33	132.64	30.20	15	127.13	27.84
Glucose	1	77.00	-	4	80.00	13.34	15	86.40	9.27	32	100.19	27.55	41	105.33	24.37	33	117.97	30.44	15	138.58	49.27
HbA1c	1	4.20	-	4	5.30	0.43	15	5.46	0.32	32	5.73	1.15	41	5.70	0.64	33	6.52	1.27	15	7.04	1.72
HOMA	1	1.90	-	4	2.55	0.21	15	3.93	2.13	32	7.51	6.44	41	9.64	6.86	33	10.44	4.63	15	15.58	8.51
CRP	1	4.80	-	4	4.70	2.36	15	4.59	2.71	32	12.01	8.89	41	11.04	6.53	33	12.42	8.30	15	18.21	11.30

**Table 3 biomedicines-11-01595-t003:** Parameter estimates for prediction of NASH. Abbreviations: BMI: body mass index; df: degrees of freedom; TG: triglyceride; HDL: high-density lipoprotein; LDL: low-density lipoprotein; HbA1c: hemoglobin A1c; HOMA: homeostasis model assessment; CRP: C-reactive protein; SD: standard deviation.

		Estimate	Std. Error				95% Confidence Interval	
				Wald	df	Sig.	Lower Bound	Upper Bound
Threshold	[NAS = 1.00]	2.917	2.186	1.781	1	0.182	−1.367	7.201
	[NAS = 2.00]	5.680	2.238	6.440	1	0.11	1.293	10.066
[aHT = 0.00]		−0.500	0.383	1.701	1	0.192	−1.251	0.251
[aHT = 1.00]		0 ^a^	-	-	-	-	-	-
BMI		0.026	0.026	1.007	1	0.316	−0.025	0.078
TG		0.003	0.0029	1.092	1	0.296	−0.003	0.009
HDL		−0.013	0.0180	0.527	1	0.468	−0.048	0.022
LDL		−0.005	0.0066	0.677	1	0.411	−0.018	0.007
Glucose		0.003	0.0128	0.060	1	0.807	−0.022	0.028
HbA1c		0.342	0.3241	1.116	1	0.291	−0.293	0.977
HOMA		0.102	0.0373	7.397	1	0.007	0.028	0.175
CRP		−0.008	0.0261	0.095	1	0.758	−0.059	0.043

Link function: Logit. The letter a is meant to indicate the reference.

**Table 4 biomedicines-11-01595-t004:** Parameter estimates for prediction of steatosis. Abbreviations: BMI: body mass index; df: degrees of freedom; TG: triglyceride; HDL: high-density lipoprotein; LDL: low-density lipoprotein; HbA1c: hemoglobin A1c; HOMA: homeostasis model assessment; CRP: C-reactive protein; SD: standard deviation; S: steatosis.

		Estimate	Std. Error				95% Confidence Interval	
				Wald	df	Sig.	Lower Bound	Upper Bound
Threshold	[S = 0.00]	4.068	2.370	2.944	1	0.086	−0.578	8.714
	[S = 1.00]	6.852	2.419	8.022	1	0.005	2.110	11.593
	[S = 2.00]	9.270	2.561	13.104	1	<0.001	4.251	14.290
[aHT = 0.00]		−0.201	0.407	0.243	1	0.622	−0.999	0.598
[aHT = 1.00]		0 ^a^	-	-	0	-	-	-
BMI		0.029	0.028	1.046	1	0.306	−0.027	0.085
TG		0.002	0.003	0.598	1	0.439	−0.004	0.008
HDL		−0.007	0.019	0.158	1	0.691	−0.044	0.029
LDL		0.002	0.007	0.047	1	0.828	−0.012	0.015
Glucose		−0.019	0.013	2.008	1	0.157	−0.045	0.007
HbA1c		0.833	0.343	5.915	1	0.015	0.162	1.505
HOMA		0.136	0.039	11.934	1	<0.001	0.059	0.214
CRP		−0.013	0.030	0.174	1	0.676	−0.072	0.046

Link function: Logit. The letter a is meant to indicate the reference.

**Table 5 biomedicines-11-01595-t005:** Parameter estimates for prediction of activity. Abbreviations: BMI: body mass index; df: degrees of freedom; TG: triglyceride; HDL: high-density lipoprotein; LDL: low-density lipoprotein; HbA1c: hemoglobin A1c; HOMA: homeostasis model assessment; CRP: C-reactive protein; SD: standard deviation; A: activity.

		Estimate	Std. Error				95% Confidence Interval	
				Wald	df	Sig.	Lower Bound	Upper Bound
Threshold	[A = 0.00]	−1.123	2.220	0.256	1	0.613	−5.474	3.227
	[A = 1.00]	0.463	2.210	0.044	1	0.834	−3.868	4.794
	[A = 2.00]	2.868	2.240	1.639	1	0.200	−1.523	7.259
	[A = 3.00]	5.548	2.459	5.089	1	0.024	0.728	10.368
[aHT = 0.00]		−0.426	0.388	1.206	1	0.272	−1.187	0.334
[aHT = 1.00]		0 ^a^	-	-	0	-	-	-
BMI		−0.017	0.027	0.399	1	0.528	−0.070	0.036
TG		0.001	0.003	0.238	1	0.625	−0.004	0.007
HDL		−0.007	0.018	0.152	1	0.696	−0.042	0.028
LDL		0.002	0.007	0.073	1	0.787	−0.011	0.015
Glucose		0.003	0.013	0.048	1	0.827	−0.022	0.028
HbA1c		0.129	0.322	0.162	1	0.687	−0.501	0.760
HOMA		0.060	0.036	2.756	1	0.097	−0.011	0.130
CRP		−0.005	0.029	0.031	1	0.859	−0.061	0.051

Link function: Logit. The letter a is meant to indicate the reference.

**Table 6 biomedicines-11-01595-t006:** Parameter estimates for prediction of fibrosis. Abbreviations: BMI: body mass index; df: degrees of freedom; TG: triglyceride; HDL: high-density lipoprotein; LDL: low-density lipoprotein; HbA1c: hemoglobin A1c; HOMA: homeostasis model assessment; CRP: C-reactive protein; SD: standard deviation; F: fibrosis.

		Estimate	Std. Error				95% Confidence Interval	
				Wald	df	Sig.	Lower Bound	Upper Bound
Threshold	[F = 0.00]	1.899	2.3396	0.659	1	0.417	−2.686	6.485
	[F = 1.00]	4.474	2.3734	3.554	1	0.059	−0.177	9.126
	[F = 2.00]	6.045	2.4494	6.090	1	0.014	1.244	10.845
[aHT = 0.00]		−0.510	0.4122	1.529	1	0.216	−1.317	0.298
[aHT = 1.00]		0 ^a^	-	-	-	-	-	-
BMI		−0.003	0.0289	0.009	1	0.925	−0.059	0.054
TG		−0.004	0.0031	1.358	1	0.244	−0.010	0.002
HDL		−0.018	0.0185	0.971	1	0.324	−0.054	0.018
LDL		0.001	0.0072	0.033	1	0.856	−0.013	0.016
Glucose		−0.019	0.0131	2.015	1	0.156	−0.044	0.007
HbA1c		1.006	0.3516	8.192	1	0.004	0.317	1.696
HOMA		0.051	0.0405	1.577	1	0.209	−0.029	0.130
CRP		0.015	0.0295	0.268	1	0.604	−0.043	0.073

Link function: Logit. The letter a is meant to indicate the reference.

## Data Availability

Data is available upon reasonable request from the corresponding author.

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
