# Peer review of "Insulin Resistance Is the Main Characteristic of Metabolically Unhealthy Obesity (MUO) Associated with NASH in Patients Undergoing Bariatric Surgery"

_biomedicines, 2023, doi:10.3390/biomedicines11061595_

Round 1

Reviewer 1 Report

Title:  Insulin resistance is the main characteristic of metabolically unhealthy obesity (MUO) associated with NASH in patients undergoing bariatric surgery.

Authors: Sophia M. Schmitz, Sebastian Storms, Alexander Koch, Christine Stier, Andreas Kroh , Karl P. Rheinwalt, Sandra Schipper, Karim Hamesch, Tom F. Ulmer, Ulf P. Neumann and Patrick H. Alizai. 

General comment:

Obesity is a very common but also very heterogeneous disease. Patients differ in their clinical presentation, including the number and type of obesity-related complications. In their study, Sophia M. Schmitz et al. attempted to assess the influence of various metabolic parameters on the risk of developing NAFLD and NASH in patients with obesity. The research hypothesis is well formulated, but there are a few methodological issues that the authors should address before the manuscript is accepted for publication.

Minor revisions:

  1. Please justify your choice of diagnostic criteria for metabolically unhealthy obesity (MUO). For example, why was CRP and not uric acid concentration included among them? Why HDL-C and triglyceride concentrations and not non-HDL cholesterol?
  2. Please provide information on whether the study participants were taking medication before surgery that may have influenced the onset and progression of hepatic steatosis - such as GLP-1 receptor agonists.
  3. Have other risk indicators for liver fibrosis such as FIB-4 been assessed in patients?
  4. Please consider the potential limitations of the study.

Author Response

General comment:

Obesity is a very common but also very heterogeneous disease. Patients differ in their clinical presentation, including the number and type of obesity-related complications. In their study, Sophia M. Schmitz et al. attempted to assess the influence of various metabolic parameters on the risk of developing NAFLD and NASH in patients with obesity. The research hypothesis is well formulated, but there are a few methodological issues that the authors should address before the manuscript is accepted for publication.

Dear Reviewer,

Thank you very much for your time to review our manuscript! We addressed all raised issues and modified the manuscript accordingly. We think your comments really added to the value of our research.

Minor revisions:

  1. Please justify your choice of diagnostic criteria for metabolically unhealthy obesity (MUO). For example, why was CRP and not uric acid concentration included among them? Why HDL-C and triglyceride concentrations and not non-HDL cholesterol?

Thank you for your comment. We agree with the reviewer, that for example uric acid concentration would be an interesting parameter to investigate in the context of MUO. However in this study we used the modified Wildman criteria as indicated in the Materials and Methods section. We clarified this by adding the references to the Introduction setting. These parameters have been established and used in other studies before which makes comparison easier.

  1. Please provide information on whether the study participants were taking medication before surgery that may have influenced the onset and progression of hepatic steatosis - such as GLP-1 receptor agonists.

This is indeed a very important comment. None of our patients received any medication for NAFLD or NASH. However, some were treated with diabetes medication that might have had an effect on onset or progression of NAFLD or NASH. Information on this was added to the manuscript. Groups were too small to test influence on NAFLD and NASH.

  1. Have other risk indicators for liver fibrosis such as FIB-4 been assessed in patients?

Thank you again for this valuable comment. We routinely assess APRI in our patients as a risk indicator for NAFLD and NASH. However, whenever possible, we rely on the liver biopsy as we did in this study.

  1. Please consider the potential limitations of the study.

The main limitation of our study is its retrospective character, that allows for description but not for assessment of mechanisms concerning the topic. However, we think our findings are an important addition to this very important topic. We added a section about the limitations of the study to the discussion section.

Reviewer 2 Report

This is a very interesting study in which liver was sampled during bariatric surgery and compared preoperative metabolic abnormalities with liver findings. Unfortunately, however, the value of this paper is severely compromised by a serious lack of information, especially in Method and Results. I strongly recommend that authors consult with a physician or specialist familiar with how to describe the original paper before reconstructing it.

-Add units to the variables in Table 1. And include age and gender in the table.

-Where can I find the definition and calculation of the NAFLD Activity Score?

-Where can I find the information of alcohol consumption in patients?

-I cannot find the subheading "Statistical Analysis" in Method. The ordinal regression model used in table 3, for example, should normally be described in detail in the Method.

None

Author Response

preoperative metabolic abnormalities with liver findings. Unfortunately, however, the value of this paper is severely compromised by a serious lack of information, especially in Method and Results. I strongly recommend that authors consult with a physician or specialist familiar with how to describe the original paper before reconstructing it.

Dear reviewer,

Thank you very much for your time invested in reviewing our manuscript. We addressed all issues and modified the manuscript accordingly. We added information on methodological background and assessment of data and clarified potential ambiguities. We also let the manuscript undergo intensive language correction with a native speaker.

-Add units to the variables in Table 1. And include age and gender in the table.

Thank you for this remark. The units have been added in Table 1. Age and sex were also included.

-Where can I find the definition and calculation of the NAFLD Activity Score?

NAFLD Activity Score was assessed by the hepato-pathologists. This was corrected in the materials and methods section.

-Where can I find the information of alcohol consumption in patients?

Patients with a history of alcohol consumption were excluded from the study. This was also added in the material and methods section.

-I cannot find the subheading "Statistical Analysis" in Method. The ordinal regression model used in table 3, for example, should normally be described in detail in the Method.

The subheading “statistical analysis” was added to the methods section. The used model was also described. We are really sorry for the inconvenience.

Reviewer 3 Report

In this study, the authors tried to assess the influence of the different subitems of metabolically unhealthy obesity (MUO) on NAFLD and NASH in patients with obesity. The authors found that the model itself was statistically significant for prediction of NASH, while the only independently significant parameter was HOMA-IR. The authors therefore hypothesize, that insulin resistance is of paramount importance in connecting NAFLD and the metabolic syndrome in a reciprocal manner.

Comments

This is an interesting study. The reviewer has some concerns as follows:

1. It is not clear whether the severity of the NAFLD and NASH in patients with obesity is related to these parameters for MUO?

2. In Table 1, the N, mean, and SD for Art. HT are lacking. It can be added.

3. Are there statistical analyses in Figures 1-4? It can be analyzed or explained.

Author Response

In this study, the authors tried to assess the influence of the different subitems of metabolically unhealthy obesity (MUO) on NAFLD and NASH in patients with obesity. The authors found that the model itself was statistically significant for prediction of NASH, while the only independently significant parameter was HOMA-IR. The authors therefore hypothesize, that insulin resistance is of paramount importance in connecting NAFLD and the metabolic syndrome in a reciprocal manner.

We would like to thank the reviewer for his or her time taken to evaluate our study and for their valuable comments. We worked on all raised issues and think that the manuscript has really improved with the help of the reviewer.

Comments

This is an interesting study. The reviewer has some concerns as follows:

  1. It is not clear whether the severity of the NAFLD and NASH in patients with obesity is related to these parameters for MUO?

Thank you very much for this remark. The research question was indeed weather these points had an impact on NAFLD and NASH. We clarified this in the introduction section.

  1. In Table 1, the N, mean, and SD for Art. HT are lacking. It can be added.

            Thank you very much for your comment. This was indeed misleading in the table, as art. HT was assessed in a dichotomous manner (positive for diagnosis for MUO or not). To clarify this issue, the table was updated.

  1. Are there statistical analyses in Figures 1-4? It can be analyzed or explained.

Thank you for your comment. We chose the analysis with ordinal regression analysis including each of the parameters counted in figures 1-4 for statistical analysis. A crosstab for example would have been another option, but we found the groups too small to be significant. We included descriptions in the figure captions to clarify the meaning.

Round 2

Reviewer 2 Report

No further comments.

No fuether comments

Reviewer 3 Report

This revised manuscript can be accepted. No further comments.